# Integrative Analysis of Gene Expression and miRNAs Reveal Biological Pathways Associated with Bud Paradormancy and Endodormancy in Grapevine

**DOI:** 10.3390/plants10040669

**Published:** 2021-03-31

**Authors:** Shuchi Smita, Michael Robben, Anup Deuja, Monica Accerbi, Pamela J. Green, Senthil Subramanian, Anne Fennell

**Affiliations:** 1Edgar McFadden BioStress Laboratory, Agronomy, Horticulture, and Plant Science Department, BioSNTR, South Dakota State University, Brookings, SD 57007, USA; shuchi2803@gmail.com (S.S.); michael.robben@sdstate.edu (M.R.); adeuja25@gmail.com (A.D.); senthil.subramanian@sdstate.edu (S.S.); 2Department of Plant and Soil Sciences and Delaware Biotechnology Institute, University of Delaware, Newark, DE 19713, USA; accerbi@dbi.udel.edu (M.A.); green@dbi.udel.edu (P.J.G.)

**Keywords:** grapevine, miRNA, photoperiod, dormancy, regulatory network

## Abstract

Transition of grapevine buds from paradormancy to endodormancy is coordinated by changes in gene expression, phytohormones, transcription factors, and other molecular regulators, but the mechanisms involved in transcriptional and post-transcriptional regulation of dormancy stages are not well delineated. To identify potential regulatory targets, an integrative analysis of differential gene expression profiles and their inverse relationships with miRNA abundance was performed in paradormant (long day (LD) 15 h) or endodormant (short day (SD), 13 h) *Vitis riparia* buds. There were 400 up- and 936 downregulated differentially expressed genes in SD relative to LD buds. Gene set and gene ontology enrichment analysis indicated that hormone signaling and cell cycling genes were downregulated in SD relative to LD buds. miRNA abundance and inverse expression analyses of miRNA target genes indicated increased abundance of miRNAs that negatively regulate genes involved with cell cycle and meristem development in endodormant buds and miRNAs targeting starch metabolism related genes in paradormant buds. Analysis of interactions between abundant miRNAs and transcription factors identified a network with coinciding regulation of cell cycle and epigenetic regulation related genes in SD buds. This network provides evidence for cross regulation occurring between miRNA and transcription factors both upstream and downstream of *MYB3R1*.

## 1. Introduction

Grapevine is a seasonally indeterminate temperate fruit crop that grows in different climates across the world. The grape industry is currently threatened by global climate change that brings with it increasingly variable temperatures in spring, fall and winter. Transition of buds from paradormancy to endodormancy synchronizes the seasonal growth cycle of the grapevine and prepares it for winter [1]. Paradormancy limits bud break in response to signals from within the vine but external to the bud maintaining its growth potential but inhibiting bud break during the growing season. Upon perception of a genotype-appropriate environmental signals, the paradormant axillary bud transitions into endodormancy, where-in bud break is controlled by factors internal to the bud [2]. Unlike many fruit tree crops, which set terminal buds to enter dormant conditions in preparation for overwintering, grapevines abscise their shoot tips and develop endodormancy in the compound axillary buds that have developed continuously throughout the growing season [3,4,5].

Physiological and transcriptomic studies in grapevine and other species reveal multiple molecular changes occur during bud endodormancy induction [3,4,5,6,7,8,9,10]. An increase in sucrose and starch metabolism and secondary metabolism related gene expression is observed during the natural temporal transition from paradormancy to endodormancy [4,7]. A comparison of microarray data from multiple species (*Euphorbia esula, Solanum tubersum*, *Rubus ideaus*, *Vitis vinifera, Arabidopsis, Populus* hybrid, *V. riparia)* show an increase in cell cycle control and circadian rhythm gene expression as common factors across the species during endodormancy induction [5,11]. In leafy spurge (*Euphorbia esula* L.) crown buds, endodormancy induction relies upon ethylene signaling and crosstalk between photoperiod and temperature response pathways [12]. During the transition from paradormancy to endodormancy in *V. vinifera* buds, an increase expression of ABA related transcripts and a decrease in the expression of gibberellin and auxin signaling pathways is observed [9]. Similarly, during photoperiod-induced endodormancy in *V. riparia* an increase in abscisic acid, raffinose, trehalose, and resveratrol concentrations occurs [5].

A classic model of endodormancy induction in apical buds of perennial plants involves photoperiod dependent reduction of phytochrome A, constans, and flowering locus T/centrorodialis-like1 (FT1/CENL1) gene expression [13,14]. This model does not explain; however, the role of transcriptional regulators and environmental factors that affect induction of endodormancy, such as, the MADS-box transcription factors or temperature dependent growth cessation [15,16]. Recent work in cherry trees, suggests that epigenetic regulation could play a vital role in dormancy transitions resulting in large-scale transcriptomic modification and an increase in expression of chromatin remodeling proteins [13,14,17].

Even though microRNAs (miRNAs) play a significant role in post-transcriptionally regulating plant developmental processes, responses to the environment, and biotic and abiotic stress during seed dormancy, few studies have looked at its role in regulating bud dormancy induction [18,19,20,21,22]. Some reports suggest that cold and drought stress induced miRNAs play an important role in grapevine cold tolerance [23,24]. A few miRNAs show a more direct role in dormancy-associated processes. Crosstalk between miR6390 and C-repeat binding factor (CBF*)*, dormancy-associated MADS-box (DAM) and flowering locus T (FT2) genes play important roles in regulating endodormancy in pear and cherry flower buds and other trees [10,17,20,22]. Similarly, both miRNA156 and miRNA172 were found to regulate seed dormancy and flowering time in lettuce (*Lactuca sativa*) and *Arabidopsis thaliana* in a delay of germination 1 (DOG1) dependent manner [21]. However, no photoperiod-inducible miRNAs with a role in regulating the dormancy induction in *Vitis* species have been reported to this date.

*V. riparia *Michx. ′Manitoba 37′. presents an excellent woody plant model to study the regulation of endodormancy induction as, in contrast to many of the tree because unlike many of the tree species models, it undergoes the dormancy transition in existing compound axillary buds rather than confounding terminal bud organogenesis and dormancy induction at the end of a growing season [7,14,25]. Previous grapevine studies in greenhouse (25 to 28 °C) indicate that 21 days of short photoperiod treatment (SD; 13 h) results in a delay of bud break with full endodormancy being induced after 28 days in SD photoperiod while the long photoperiod (LD; 15 h) treated grapevines remain paradormant [26]. Co-incident with bud endodormancy induction a greater number of downregulated genes were observed during the endodormancy phase starting at 28 days of SD [5]. Therefore, in the present study, RNA-seq was used to analyze differential gene expression at this pivotal stage and miRNA abundance and inverse gene expression analysis and predicted motif gene set enrichment of downregulated genes were used to identify potential regulatory networks associated with bud endodormancy.

## 2. Results

### 2.1. Endodormancy Is Associated with Greater Down-Regulation of Gene Expression Relative to Paradormancy

Hierarchal clustering indicated the transcriptome profiles of the paradormant (LD) and endodormant (SD) buds are distinctly different (Appendix A, Appendix A). There were 1336 genes differentially expressed (DEG) in the buds, of these, 400 were upregulated and 936 were downregulated in the SD buds relative to the LD buds (Appendix A). A clear distinction between the two treatments is observed (Appendix A), but there was variability between replications within each photoperiod treatment. Therefore, the DEG identified in this study were compared to the DEG in a previously published microarray study with similar conditions, resulting a strong correlation (R^2^ 0.8, Appendix A, Appendix A). An overview of changes in biological processes and molecular function was obtained using gene set enrichment analysis (GSEA) with Vitis Network annotation. GSEA results using all expressed genes showed 26 VitisNet pathways were positively enriched in SD, notably glycolysis, phenylpropanoid biosynthesis, and starch and sucrose biosynthesis (Appendix A). There were no transcription factor pathways enriched in the SD buds; however, there were 19 transcription factor pathways enriched in LD buds. In addition, hormone signaling pathways were positively enriched in LD relative to SD buds. Gene ontology (GO) analysis of the differentially expressed genes was conducted to identify specific gene function and metabolic pathway enrichments in SD and LD treatments. GO enrichment analysis of the up-regulated genes in SD relative to LD buds identified 10 biological process, two cellular component, and 24 molecular function categories (Figure 1, Appendix A). In contrast, there were 136 biological processes, 19 cellular component, and 10 molecular function categories enriched for the down-regulated genes in SD relative to the LD buds. In agreement with the GSEA overview, the enriched GO for biological processes of upregulated genes in SD buds were in oxidation-reduction, phenylpropanoid, lignin, starch catabolic and stress related categories. Molecular function categories for upregulated genes in SD were enriched in oxido-reductase activity and catalytic activity. In contrast, biological processes enriched for the genes downregulated in SD were signaling, cell cycle, chromatin organization, and shoot and floral development related categories.

### 2.2. miRNA Abundance Differs in Paradormant (LD) and Endocormant (SD) Buds

A BLASTN search of SD and LD sequenced small RNAs against the miRBASE repository revealed that 62,612 of the unique small RNA sequences matched with 2626 conserved (validated sequences in any species) and known miRNAs (validated sequences specific to *V. vinifera*) belonging to 1987 miRNAs families, from at least 61 plant species (out of 73 Viridiplantae species in miRBASE) (Appendix A, Appendix A). A set of 2094 unique sequences (corresponding to 339 conserved miRNAs; 224 miR families; in 48 plant species) were classified as highly abundant (≥ 20 counts in both libraries combined) (Appendix A). There were 252 conserved miRNAs with ≥ 2-fold difference in abundance between LD and SD buds, 139 (belonging to 92 miR families) and 113 (belonging to 87 miR families) had a greater abundance in LD and SD, respectively (Table 1). Some miRNAs in the LD (miRNA166m, miRNA3636) and SD (miRNA397b, miR408b, miR156s) had greater than 20-fold change in abundance between the two treatments (Appendix A, Appendix A). The differential abundance of 13 LD and 13 SD abundant miRNAs was validated by qPCR and coinciding with the small RNA sequence abundance, nine of the LD and three of the SD were significantly abundant (Appendix A). A regression of the mean log fold change (FC) qPCR values with small RNA-seq log FC data yielded an R^2^ value of 0.6 with some of the LD miRNAs showing greater FC. A total of 50 conserved and 11 known miR families were exclusively downregulated during endodormancy induction while 47 conserved and 9 known miR families were exclusively upregulated. Seventeen of these families were significantly differentially expressed with a Wilcoxon *p*-value < 0.05 and all but one had a fold change greater than two (Table 1).

### 2.3. miRNAs Target Different Genes in Paradormant (LD) and Endodormant (SD) Buds

Gene target prediction of the LD and SD abundant miRNAs was conducted to elucidate pathways that the miRNAs might regulate during dormancy. A total of 113 LD abundant miRNAs had 1571 potential predicted target genes and 88 SD abundant miRNAs had 1290 potential predicted target genes (Appendix A). Many members of the miR166 family, which were abundant in LD buds were predicted to target thaumatin genes, a non-conserved target gene for this miRNA family. Squamosa promoter binding protein-like 4 (SPL4) was predicted to be targeted by miRNA156 (a, b, I, l, s isoforms) abundant in SD buds. GO enrichment analysis of predicted target genes revealed that the biological processes such as post-transcriptional regulation of gene expression and silencing by RNA related genes were enriched in both LD and SD (Appendix A, Appendix A). Predicted gene targets of LD abundant miRNAs were uniquely enriched in defense response, response to stress, and stimulus related biological processes. In contrast, predicted target genes of the SD abundant miRNAs were uniquely enriched for biological processes related to lignin metabolism and catabolism, aromatic compound catabolism, reproductive system and structure development, and embryo development ending in dormancy (Figure 2, Appendix A).

### 2.4. Inverse Expression Association of Predicted Target Genes and Abundant miRNAs

Using inverse expression analysis of the miRNAs and their predicted target genes that were differentially expressed, 12 predicted target genes of 14 LD abundant miRNAs and 31 predicted target genes of 26 SD abundant miRNAs were identified as inversely correlated in LD and SD buds, respectively (Figure 3a,b; Appendix A). LD abundant miRNAs (miR166a, +miR772, miR1863, and miR159) were predicted to target thaumatin, trehalose-phosphatase, and isoamylase genes which showed decreased expression levels in LD buds (Appendix A). Most of the LD abundant miRNAs predicted target genes were a part of the starch metabolism or pathogen resistance pathways (Figure 3a, Appendix A).

An inverse expression relationship was found between expression of a squamosal promoter-binding-like protein 4 (SPL4) gene and miR156 in SD buds, with one of the miR156 also predicted to target a basic helix-loop-helix transcription factor (BHLH) (Figure 3b, Appendix A). Two ethylene response genes, whose expression was significantly downregulated in SD relative to LD were the predicted targets of the SD abundant miR3633. Interestingly, BHLH, peroxidase, glycoprotein 11, and ferulated 5-hydroxylase genes all had a similar decreased expression pattern in SD buds relative to LD buds from previously published microarray data from our lab. These observations suggest potential post-transcriptional regulation of these genes by the respective miRNAs during dormancy transition.

### 2.5. Predition of Transcription Factor Regulatory Network in LD and SD Abundant miRNAs

Using predicted motif enrichment of transcription factor binding sites, we performed a GSEA of differentially expressed genes using predicted transcription factors as gene sets and found MYB3R1 transcription factor to be the only transcription factor gene set significantly enriched for downregulated genes in the SD. In the SD buds, downregulated genes with MYB3R1 binding motifs in their promoters were involved in pathways belonging to cell cycle, cytoskeleton, extracellular, and vesicular transport, and various other transcription factors. To understand how miRNA and transcriptional regulators might work together in the endodormant bud a network was constructed with transcriptional targets and protein interactors of MYB3R1, as well as miRNAs and their interactions with their predicted targets (Figure 4, Appendix A). We observed key genes involved in pathways such as the cell cycle and epigenetic modification pathways that were upregulated (ie protein interactors maintenance of methylation (MOM1) and retinoblastoma related 1 (RBR1_1)), or downregulated (cyclins (CYC) and mitotic spindle checkpoint protein (MAD2)), in endodormant buds (Figure 4, Appendix A). This network suggests cross regulation between miRNA and transcription factors both upstream and downstream of MYB3R1 (Appendix A). In particular, the transcription factors auxin regulation factor 9 (ARF2) and serine/threonine protein kinase aurora (AURORA1) were predicted targets of miR397 and miR156, respectively. Six miRNA families (miR156, 167, 397, 1863, 5139, 8155, 8556) abundant in SD buds were predicted to target members of MYB3R1 network including chromomethylase (CMT), minichromosome maintenance proteins (MCM2,3,4,6), chinese for ugly (TSO1), kinesin, kinesin motor protein (KMOTOR) and AURORA-1 (AURORA1). These miRNAs all target transcription factors or protein interactors that are members of a larger MYB3R1 cell cycle regulatory hub.

## 3. Discussion

Transcription factors and miRNAs have emerged as major drivers of transcriptional and post-transcriptional regulation during plant development and growth. The large-scale sequencing of transcriptomes has been a key instrument in identifying the effect of this regulation in a variety of non-model plant systems [27]. RNA-seq has also been used to explore gene expression of miRNAs and other regulatory elements [28]. Tissue expression profiles of miRNAs has identified important regulators in *V. vinifera*, including those expressed in the dormant bud [29,30,31,32]. While RNA-seq has been applied to bud endodormancy in *V. vinifera* [33], gene expression during the shift to endodormancy in *V. riparia* has only been studied using microarray technology [3,5].

In this study, we integrated mRNA and miRNA RNA-seq data to deduce potential regulators in paradormancy maintenance and endodormancy induction. RNA-seq analysis of LD or SD treated *V. riparia* buds revealed 1336 genes differentially regulated between the two treatments. The majority were downregulated in SD relative to LD buds, suggesting transcriptional regulation may be associated with the shift to endodormancy. Genes upregulated upon entering endodormancy are involved in metabolic pathways, phenylpropanoid synthesis, membrane transport, glycolysis, and sucrose and starch metabolism. These results are notable since starch concentrations were found to be high at the onset of endodormancy in the xylem sap of Japanese Pear [34], and trehalose and raffinose production increase to a peak level after 42 days of SD treatment in *V. riparia* [5]. Genes downregulated in association with endodormancy were hormone signaling, transcription factors, cell cycle, and chromatin organization.

Small RNA sequencing data analysis and mapping with miRBASE demonstrated varied abundance of miRNAs among families. There was a lower abundance of annotated miRNAs in endodormant buds, which could be due to the annotated set of our small RNA library containing regulatory elements and also non-regulatory small RNAs and the remains of mRNA degradation. This is supported by the fact that miRNAs upregulated in endodormant conditions have more predicted targets as well as more cases of inverse gene expression with the mRNA data. miR families 166 and 167 were the most significantly upregulated families in paradormant grapevine buds, which is consistent with studies in pear winter flower buds [22].

Using inverse patterns of mRNA and miRNA expression, regulators of both paradormant and endodormant conditions were identified. Most notably, the miRNAs abundance coincided with reduced starch and sucrose metabolism gene expression in LD buds. There is also an inverse relationship between miR159 abundance and the trehalose-6-phosphatase gene expression level in LD but not SD buds. The predicted downregulation of trehalose-6phosphatase having a role in promoting paradormancy maintenance is supported by the contrasting enhanced trehalose-6-phosphatase gene expression during bud endodormancy in *V. riparia* [5]. In addition, trehalose-6-phosphate content is also correlated with bud outgrowth potential in potatoes and peas [31,32,35,36,37] Similarly, miR3624 which was predicted to target heavy metal domain associated protein genes, that have been shown to produce reactive oxygen species, was abundant in paradormant buds in contrast to endodormant buds. In the grapevine rootstock M4, drought stress inhibited miR3624 expression and the heavy metal associated gene is upregulated [24]. Similarly, heavy metal domain associated protein genes were upregulated in endodormant buds in contrast to paradormant buds in this study. Thus, the increased abundance of miRNA3624 in paradormant buds may play a role in limiting active oxygen species production and maintaining paradormancy.

Genes that were predicted to be regulated by miRNAs in SD were related to cell cycle and cell division pathways. This is in agreement with previous findings that endodormancy is associated with the cessation of vegetative, and flower meristem development, while sustained paradormancy is associated with the development of multiple branched floral primordia [38]. The miR156 family of miRNAs, which has been shown to control the meristem cell fate transition in many plants, was prominent in these results [22]. An increased abundance of miR156 was inversely correlated with decreased gene expression of two floral related genes (squamosa-promoter binding protein-like transcription factor gene (SPL4) and unusual floral organ (UFO)) in endodormant buds. In lettuce, reduced miR156 promoted early flowering and seed germination; in contrast, increased miR156 increased seed dormancy [21]. Downregulation of SPL4 by miR156 in switchgrass (*Panicum virgatum*) SPL4 promoted axillary bud initiation [39] In grapevine, reduced expression of SPL family genes was correlated with endodormancy [4,5]. These imply that miR156 has a strong role in regulating bud dormancy. Another, miRNA482, which we predicted to target and reduce expression of kinesins, has been found to directly impact soybean nodulation, a highly active cell division stage of development [40]. In addition, ethylene-signaling genes were targeted by multiple miRNAs which are highly abundant in the endodormant buds. In conjunction with the findings that most ethylene pathway genes were downregulated we speculate that miRNA negative regulation of the ethylene signaling pathway plays an important role in photoperiod induced endodormancy. In endodormant buds an inverse relation was identified between miR397 abundance and laccase gene expression. Similarly, a marked decrease in laccase and other cellulose synthase genes found during Thompson Seedless (*V. vinifera*) bud dormancy [41]. The regulation of laccase, which has a role in lignin biosynthesis, is well documented in plants; however, whether it has a role in dormancy regulation is not known [42,43].

Transcription factor regulation is an important component of transition into endodormancy in many species and it is widely accepted that DAM transcription factors play a vital role in the process in pear and apricot buds [13,18,20]. There were increased miRNA abundances and an inverse expression of several transcription factor predicted targets, including BHLH, SPL4, and ERF105 upon entering endodormancy in *V. riparia* buds (Appendix A). MYB3R1 motifs were recently found enriched in differentially expressed genes during hypoxia-related bud break in *V. vinifera* [44]. Similarly, MYB3R1 was the only transcription factor to have motifs significantly enriched in the promoters of downregulated genes in *V. riparia* endodormant buds. While MYB3R1 is not differentially expressed in our data set, upstream protein interactors including LRR-repeat receptor kinases, cell-cycle control genes, and kinesin motor proteins were predominantly downregulated excluding checkpoint proteins such as the plant homolog retinoblastoma related 1 (RBR1) and an ABA insensitive 1 related protein phosphatase 2C (DBPP2C). Likewise, it is noted in pea that transition of buds from dormant to active growth show results in high levels of phosphorylated RBR1 [45]. Expression of some cell cycle genes such as AURORA1, MAD2, and kinesin appear simultaneously suppressed by MYB3R1 and miRNAs 156, 167, and 1863. This leads us to believe that a signal transduction and miRNA dependent pathway might regulate the activity of MYB3R1 to suppress cell division in preparation for endodormancy. It is noted in *Arabidopsis,* that MYB3R1, 3, and 5 gene expression is required to suppress cell cycle genes after DNA damage [46]. This network also pointed to the interaction of MYB3R1 with histones and various epigenetic proteins which has not been previously shown in other studies. In particular, the increased expression of MOM1 and decreased expression of decreased DNA methylation 1 (DDM1) genes which could possibly further regulate MYB3R1-dependent gene expression by methylation of DNA or histone H3.

This study like previous grapevine dormancy studies had a greater number of genes downregulated than upregulated upon transitioning to bud endodormancy in grapevine, supporting a potential role for miRNA in transcriptional regulation. Inverse analysis and predicted motif gene set enrichment of downregulated genes identified potential miRNA and transcriptional regulatory networks associated with bud endodormancy. A potential role in paradormancy maintenance was identified for miR159 regulation of trehalose-6-phosphatase and miR3624 regulation of heavy metal domain associated protein genes. In endodormancy, a regulatory role is suggested for miR156 and miR396 through downregulation of floral and meristem development related genes. The predicted motif gene set enrichment in downregulated genes identified potential regulatory networks associated with the MYB3R1 transcription factor. This network identified potential cross regulation between miRNA (notably 156, 167 and 1863) and transcription factors up- and downstream of MYB3R1 and involved cell cycle and epigenetic pathways in endodormant buds.

## 4. Materials and Methods

### 4.1. Plant Materials and Photoperiod Treatments

Six-year-old spur pruned ecodormant *V. riparia* ‘Manitoba 37′ Michx. (PI588259)) vines were repotted and grown in long photoperiod (LD, 15 h) at 25/20 ± 3 °C day/night temperatures with 600–1400 mol m^−2^ s^−1^ photosynthetic photon flux in a climate-controlled unshaded glass greenhouse (EnTech Control Systems Inc., Montrose, MN, USA) in Brookings, South Dakota (44.3 N). Grapevines were grown for 30 days post bud break reaching shoot lengths of 10–15 nodes. Three replicate ten-vine experimental units were randomly assigned to each photoperiod treatment of continued LD (paradormancy) or short photoperiod (SD, 13 h; endodormancy) as previously described [5]. After 28 days, three replicate bud samples for each photoperiod were harvested into separate tubes of liquid nitrogen and stored at −80 °C until small RNA or total RNA extraction.

### 4.2. RNA Sequencing and Differential Gene Expression Analysis

Total RNA was extracted from three biological replicates of LD and SD buds using a modified method of Chang et al. [47] as described in Mathiason et al. [3]. DNA free RNA quality and quantity were verified with an Agilent (Santa Clara, CA, USA) 2100 Bioanalyzer RNA 6000 nano chip. RNA-seq libraries for LD and SD bud transcriptomes were prepared and sequenced by Illumina HiScanSQ (100 bp, single strand) at the Cornell University Institute of Biotechnology Genome Facility (Ithaca, NY, USA). The full raw sequencing data were submitted to the GEO database with the accession number GSE95429. Raw read quality and adapter sequences were checked using Fastqc (v. 0.11.3) (http://www.bioinformatics.babraham.ac.uk/projects/fastqc/, accessed on 10 October 2020). Further adapter trimming and filtering of poor quality sequences were performed using Cutadapt (v. 1.8.1) and Prinseq-lite (v0.20.4) (minimum phred quality score = 20, minimum post trimming length = 20), respectively [48]. The reference genome based alignment and assembly of clean reads were performed using Tuxedo pipeline [49]. The V. vinifera PN40024 12X V2 genome was indexed using Bowtie2 (v.2.2.4). Quality reads were aligned using TopHat (v2.0.13) to the indexed reference genome and guided using gene models available for *V. vinifera* (http://genomes.cribi.unipd.it/DATA/GFF/F17, accessed on 4 October 2019). Mapped reads in each library were assembled into transcripts using cufflinks (v. 2.2.1). Assembled transcript mapping to the reference genome, identified a total of 29,288 genes expressed at a threshold of FPKM ≥ 1 in at least one of the dormancy phases. Principal component analysis indicated that the treatment replicates separated into LD and SD groups. Differential gene expression (LD vs. SD) was analyzed using cuffdiff (v. 2.2.1) with adjusted *p*-value ≤ 0.05 and fold change (≥ or ≤1). *V. vinifera* PN40024 12X_V1 and _V3 gene annotation and VitisNet annotation were used to identify the significant genes [50]. Differentially expressed genes from a microarray study with similar photoperiod treatments [5] was utilized to calculate correlation with current RNseq data using “cor.test” function in R.

### 4.3. Transcriptome Functional Enrichment Analyses

Functional enrichment analysis was performed by the g:Profiler (http://biit.cs.ut.ee/gprofiler/ accessed on 10 October 2020) using the hypergeometric distribution adjusted by set count sizes (SCS) for multiple hypothesis correction [51]. SCS threshold removes enriched false positive GO terms and prioritizes truly significant results. Each gene was assigned a GO term, if it crossed the threshold adjusted *p*-value (SCS) ≤ 0.05. GO annotation of Vitis genes was obtained from the *V. vinifera* 12X_V1 genome annotation as background.

### 4.4. VitisNet Gene Set Enrichment Analysis (GSEA)

Gene Set Enrichment Analysis (GSEA) was conducted using read count data from each of the three replicates for each treatment using GSEA-P 2.0 (http://www.broad.mit.edu/GSEA accessed on 10 October 2020) [52] and 247 VitisNet pathways (https://openprairie.sdstate.edu/vitisnet-12x_files/ accessed on 10 October 2020) including at least 7 genes. The recommended GSEA-P 2.0 default parameters of 1000 permutations, nominal *p*-value < 0.05 and FDR q-value < 0.25 were used to identify enriched molecular pathways related to LD or SD. Positive significantly enriched pathways are generally upregulated in SD bud relative to LD bud.

### 4.5. Small RNA Library Construction, Sequencing and Processing

Total RNA was isolated using Plant RNA Reagent (Invitrogen/Thermo Fisher Scientific, Carlsbad, CA, USA), and low molecular weight RNA was purified from total RNA by PEG8000/NaCl precipitation [53,54]. Two small RNA libraries were constructed: (1) a pool of LD and (2) a pool of SD buds as described [55] with the following: RNA oligos for RNA ligation:5′ RNA adapter: 5′-GUUCAGAGUUCUACAGUCCGACGAUC-3′, 3′ RNA adapter: 5′-pUCGUAUGCCGUCUUCUGCUUG-idT-3′ (p, phosphate; idT, inverted deoxythymidine); DNA oligo for reverse transcription: RT-primer (5′-CAAGCAGAAGACGGCATACGA-3′); DNA oligos for PCR amplification:5′ PCR primer (5′-AATGATACGGCGACCACCGACAGGTTCAGAGTTCTACAGTCCGA-3′), 3′ PCR primer (5′-CAAGCAGAAGACGGCATACGA-3′). The Illumina sequencing of small RNA libraries from LD and SD led to the generation of 7,761,608 and 6,530,774 non redundant sequences, respectively. Filtering the low quality reads and adaptor contaminations resulted in 7,591,740 (LD) (total read count 24,452,821) and 6,352,594 (SD) (total read count 22,206,946) unique sequences that were retained for further analysis.

High quality, non-redundant small RNA sequences were aligned with the miRNA precursor/mature miRNAs of all Viridiplantae (green plants) in miRBase 21.0 (http://www.mirbase.org/ accessed on 4 October 2019) to identify annotated (conserved and known in *V. vinifera* (vvi) miRNAs in *V. riparia*. Utilizing Blastn, sequences with ≤2 mismatches were considered as potential miRNAs. Sequences matched with already deposited *V. vinifera* miRNAs were designated as “known in vvi” and sequences that matched to other plant species were designated as “conserved miRNAs”. The total conserved and miRNAs sequences known in vvi with ≥ 20 counts total in the LD and SD libraries were called “highly abundant” miRNAs and were used for further analyses. Reads per million (RPM) value was calculated for each identified miRNAs sequence (normalized expression = number of reads for a sequence/number of total reads for all sequences in that library × 1,000,000). The differential abundance of miRNAs sequences between LD and SD was calculated as fold-change = log_2_ (miRNAs normalized RPM in ‘LD’/normalized RPM in ‘SD’). miR families abundant (≥2 fold), only in LD or SD library were called “specifically abundant” miR families. Using a Wilcoxon Ranked Sum test, total read counts of miRNAs that annotated to specific miRNA families were used to calculate a *p*-value statistic between LD and SD expression in each family.

Quantitative PCR validation was conducted for 26 miRNAs, 13 each from LD and SD abundant miRNAs from 18 miRNA families. cDNA was produced for each of three biological replicates using 500 ng of RNA as described by Turner et al. [56]. cDNA and gene-specific primers (Appendix A) were combined with 2X SYBR Master Mix and ROX baseline dye (Thermofisher Scientific, Waltham, MA, USA) and qPCR was performed for three technical replicates of each the three biological replicates of cDNA using the running conditions 95 °C for 5 min, followed by 45 cycles of 95 °C for 5 s, 60 °C for 10 s and 72 °C for 1 s. ΔCt of each miRNA was calculated against the U6 control in each pooled cDNA group using the equation (ΔCt = Ct_Control_ − Ct_Gene)_) and the ΔΔCt was calculated using the equation (ΔΔCt = ΔCt_LD_ − ΔCt_SD_). Log fold change values were determined and statistical analyses for pairwise comparison of the ΔCt values were by Student’s *t*-test, *p*-value < 0.05 for the LD and SD treatments.

### 4.6. Target Prediction for Abundant miRNAs

Empirical parameters were used with an in-house perl script to run Patscan (http://blog.theseed.org/servers/2010/07/scan-for-matches.html accessed on 4 October 2019) and RNAduplex [57] for recognition of potential targets of abundant miRNAs using *V. vinifera* as reference mRNAs. The empirically inferred parameters tuned with maximum one mismatch at position 2–9, no mismatch at position 10–11 and 4 mismatch from position 12 to end [58,59]. The output was parsed to identify hits on complementary strand with <3 consecutive mismatches and relative minimum free energy (MFE) ≥70% compared to perfectly complementary target genes. Cytoscape software was used to visualize the miRNA-target regulatory networks [60].

### 4.7. Transcription Factor Motif GSEA and Network

*V. vinifera* specific transcription factor, regulated gene sets from enriched motif predictions in TFDB (downloaded 12/10/2018) were used in a GSEA analysis through the gage package [61]. Predictions of protein interactions in *V. vinifera* were obtained from BIOgrid and limited to interactions with a confidence score greater than 700. Protein interactions combined with Protein-DNA interactions and miRNA-target predictions were constructed into a network using Cytoscape [60] complimented with RNA-seq log fold change scores.

## Figures and Tables

**Figure 1 plants-10-00669-f001:**
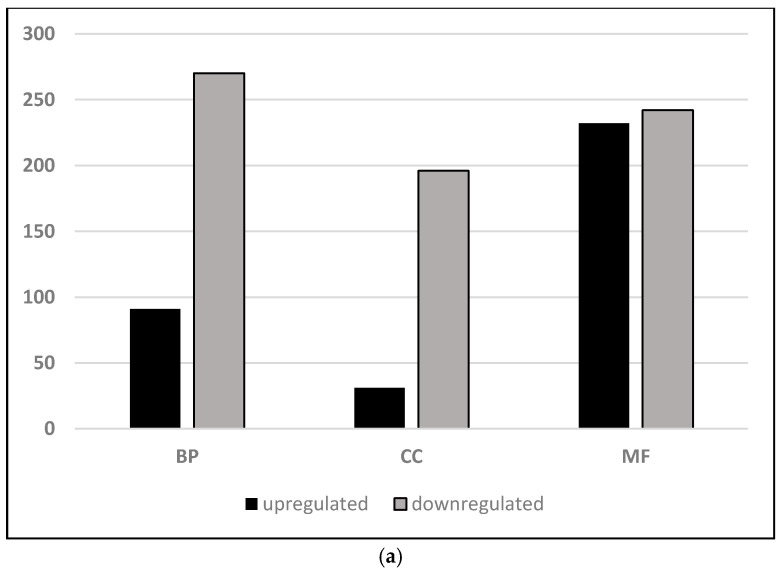
(**a**). Biological processes (BP), cellular component (CC), and molecular function (MF) gene ontology categorization of differentially expressed genes in *V. riparia* buds. Bars represent the gene ontology of non-redundant differentially expressed genes upregulated (black) and downregulated (gray) in SD relative to LD buds. (**b**) Molecular function categories enriched in upregulated (black) and down (gray) differentially expressed genes.

**Figure 2 plants-10-00669-f002:**
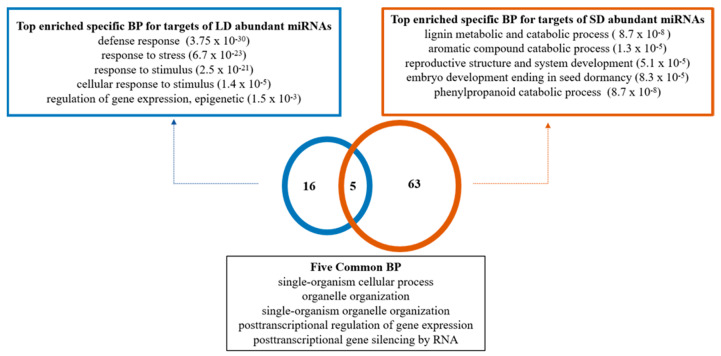
Biological processes (BP) enrichment of predicted targets for abundant miRNAs in *V. riparia* buds. Blue outlines indicate paradormancy (LD, 15 h) and orange outlines indicate endodormancy (SD, 13 h) related enriched BP for targets. Significance of BP enrichment was calculated using hypergeometric distribution adjusted by set count sizes (SCS) for multiple hypothesis correction. Adjusted *p*-value is indicated in parenthesis after each biological process GO term.

**Figure 3 plants-10-00669-f003:**
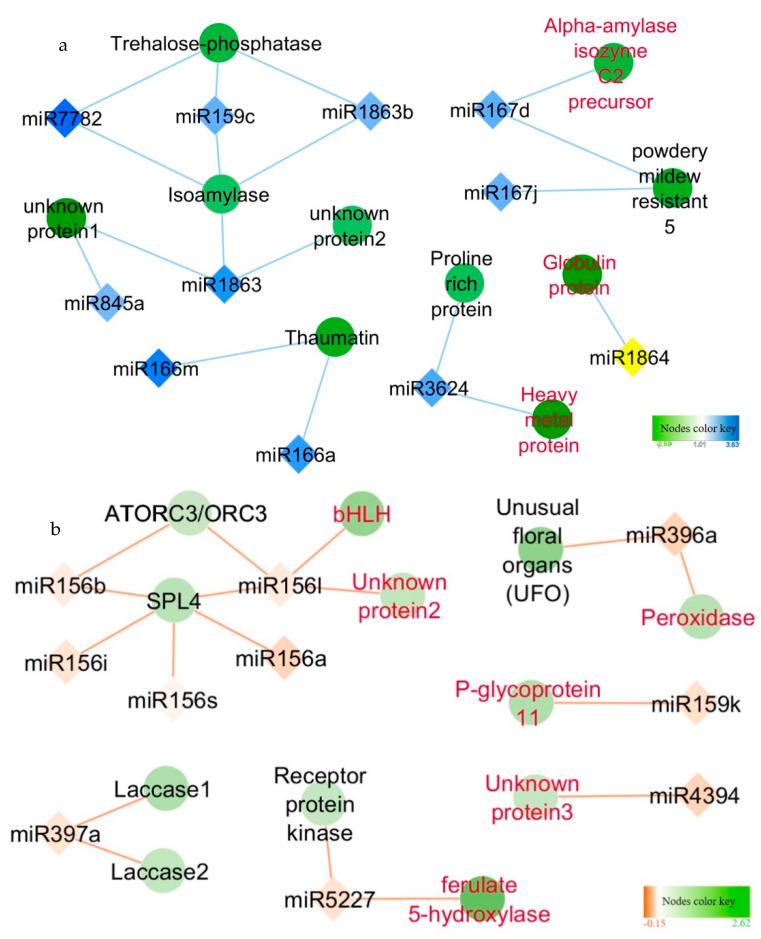
Example portions of miRNA-mediated (**a**) Stress networks for paradormant buds (LD, 15 h). (**b**) Meristem regulatory networks for endodormant buds (SD, 13 h). in *V. riparia* buds. Nodes are displayed for LD (**a**, blue diamonds) and SD (**b**, orange diamonds) abundant miRNAs, their target genes (green circle). The intensity of the node color is directly proportional to their expression. The complete networks of the inverse pattern of target genes are identified in Appendix A. Genes identified in red font are genes that were noted to be downregulated in multiple dormancy studies (Appendix A).

**Figure 4 plants-10-00669-f004:**
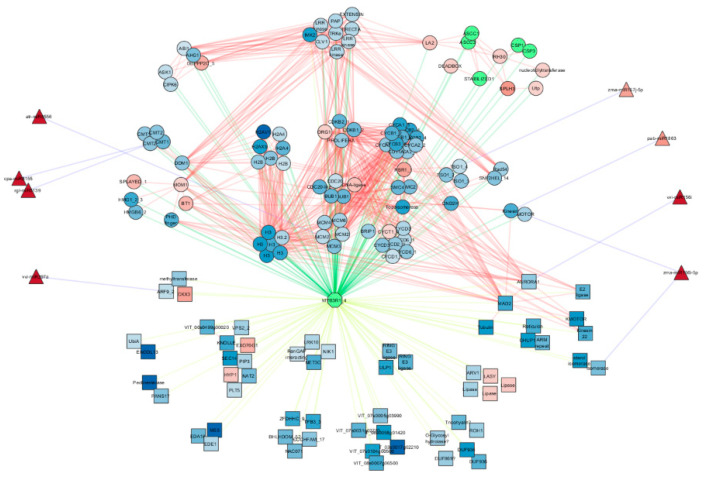
Protein network of MYB3R1 interaction with transcriptional targets (squares), protein interactors (circles), and miRNA (triangles). Transcriptional target, protein interactors, and miRNA nodes are represented by squares, circles, and triangles, respectively. MYB3R1-target interactions are denoted by yellow edges, MYB3R1-protein interactions by green edges, protein-protein interactions by red edges, and miRNA-target by blue edges. The node fill color (blue (negative) to red (positive)) indicates the log fold change expression values for differentially expressed genes and nodes with green fill color were not differentially expressed in SD relative to LD buds (Appendix A).

**Table 1 plants-10-00669-t001:** Significant enrichment of miRNA families during paradormancy (LD, 15 h) and endodormancy induction (SD, 13 h) in *V. riparia* buds.

miR Family	Fold (LD vs. SD)	FDR Adjusted *p*-Value	Mean LD Count	Mean SD Count	miRNA Sequences with ≥20	miR Family Logo ^1^
166	2.7	9.2 × 10^−10^	604	225	431	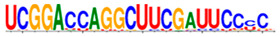
**167**	**2.5**	**6.1 × 10^−13^**	**2933**	**1164**	**170**	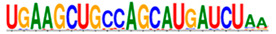
894	4.1	4.3 × 10^−5^	237	58	64	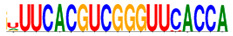
**3636**	**4.7**	**4.1 × 10^−7^**	**929**	**197**	**45**	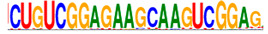
**3623**	**3.1**	**7.0 × 10^−4^**	**629**	**201**	**42**	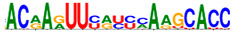
**169**	**3.2**	**2.6 × 10^−2^**	**264**	**82**	**27**	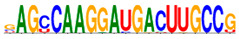
390	3.0	3.4 × 10^−2^	936	309	20	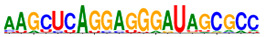
**3640**	**3.6**	**1.4 × 10^−2^**	**207**	**58**	**19**	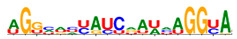
**3637**	**3.0**	**6.6 × 10^−3^**	**145**	**48**	**16**	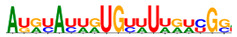
164	2.5	1.1 × 10^−2^	25	10	14	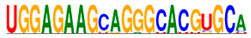
391	3.0	2.7 × 10^−2^	563	185	12	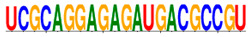
2916	−2.3	3.5 × 10^−6^	53	121	96	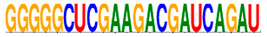
**3639**	**−2.0**	**2.7 × 10^−2^**	**225**	**445**	**45**	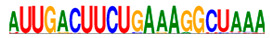
**156**	**−3.6**	**1.8 ×10^−2^**	**348**	**1254**	**24**	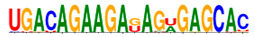
**397**	**−3.4**	**1.3 × 10^−2^**	**347**	**1168**	**16**	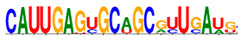
**408**	**−3.8**	**3.6 × 10^−2^**	**76**	**289**	**15**	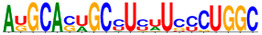
**398**	**−3.9**	**1.2 × 10^−2^**	**141**	**545**	**10**	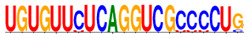

^1.^ Denotes the miR family HMM logo for the sequences having ≥20 counts in library. miR families in bold are reported with known vvi miRNA member. (For all miRNAs see Appendix A).

## Data Availability

The RNA-seq data has been deposited in NCBI under accession number GSE95707.; The small RNA data has been deposited at http://openprairie.sdstate.edu/vitis_riparia_microRNA_data, accessed on 4 March 2021.

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
