# Peer review of "Integrative Analysis of Gene Expression and miRNAs Reveal Biological Pathways Associated with Bud Paradormancy and Endodormancy in Grapevine"

_plants, 2021, doi:10.3390/plants10040669_

Round 1
Reviewer 1 Report
Lines 48-50: sentence fragment or incomplete thought.
Line 51: different species is unclear until the next sentence. In this sentence, it could also refer to different Vitus species.
Line 53: both citations 11 and 12 refer to leafy spurge. Perhaps include citation 12 here.
Line 56: this sentence refers to grape. The authors need to be clearer about which system they're discussing. Also, this citation could also be included in the citations listed in Line 53.
Line 60: please consider replacing the "/" with "and". Even though most readers will realize that PHYA and CONSTANS are different genes, the later use of "/" in the sentence properly denotes different nomenclatures for the same gene.
Line 88: please include the temperature(s) at which this study was done.
Figure 2: data suggest that the biological samples were not consistent. If true, there is no acknowledgment of this in the text.
Line 378: suggest "Six-year old spur pruned..."
Line 391, Mathiason reference: perhaps the authors meant reference #3? Reference 35 is completely different.
Author Response
Response to Reviewer 1
Thank you for taking the time to provide comments on the manuscript. Please see point-by-point responses below.
Lines 48-50: sentence fragment or incomplete thought.
This sentence has been completed: An increase in sucrose and starch metabolism and secondary metabolism related gene expression is observed during the natural temporal transition from paradormancy to endodormancy [4,7].
Line 51: different species is unclear until the next sentence. In this sentence, it could also refer to different Vitus species.
The sentence originally including “different species” has been changed and species added in parenthesis: A comparison of microarray data from multiple species (Euphorbia esula, Solanum tubersum, Rubus ideaus, Vitis vinifera, Arabidopsis, Populus hybrid, V. riparia) show an increase in cell cycle control and circadian rhythm gene expression as common factors across the species during endodormancy induction [5,11].
Line 53: both citations 11 and 12 refer to leafy spurge. Perhaps include citation 12 here.
The sentence for lines 50 to 53 is specifically identifying microarray data that is compared across multiple species. The sentence in line 53 to 55 is Leafy spurge only and no comparison is made across the total DEG to multiple species as referenced in the previous sentence. Thus, the citations are accurate with the statements and citation #12 should not be included with lines 50 to 53.
Line 56: this sentence refers to grape. The authors need to be clearer about which system they're discussing. Also, this citation could also be included in the citations listed in Line 53.
This has been clarified: During the transition from paradormancy to endodormancy in V. vinifera buds, an increase expression of ABA related transcripts and a decrease in the expression of gibberellin and auxin signaling pathways is observed. Citation #9 is not used in line 53 as that sentences is specific to making large scale transcriptome comparisons across species. Citation #9 does not make such comparisons.
Line 60: please consider replacing the "/" with "and". Even though most readers will realize that PHYA and CONSTANS are different genes, the later use of "/" in the sentence properly denotes different nomenclatures for the same gene.
This has been corrected.
Line 88: please include the temperature(s) at which this study was done.
This has been included (25 to 28ÌŠ C)
Figure 2: data suggest that the biological samples were not consistent. If true, there is no acknowledgment of this in the text.
Yes there is variation in the replicates. A statement is inserted in the results and RNAseq differentially expressed genes have been compared to previously published microarray data that had similar photoperiod treatments. There was a strong correlation (R2 of 0.8) between the DEGs of the microarray and RNAseq studies conducted in different years. A new figure (Figure S2) has been added to the results for this comparison. The FPKM for all replicates have been inserted in Table S1. Figure 2 has been removed as the statistically significant positive enrichment of the hormone signaling genes are found in the Gene Set Enrichment Analysis (GSEA, Table S2) and the figures 1a and 1b summarize the transciptomic profiles. It is important to note we report systematic changes in gene expression across pathways in the results and are not relying on specific striking difference in gene expression.
Line 378: suggest "Six-year old spur pruned..."
This has been changed to: Six-year old spur pruned ecodormant V. riparia (Michx. (‘Manitoba 37’ PI588259)) vines were repotted
Line 391, Mathiason reference: perhaps the authors meant reference #3? Reference 35 is completely different.
You are correct, this has been changed to citation #3.
Reviewer 2 Report
The manuscript by Smita and colleagues presents the grapevine (Vitis riparia) RNAseq transcriptome from long day (15h) paradormant buds and short day (13h) endodormant buds. A detailed bioninformatic analysis is presented characterizing the differentially expression bud transcriptome and some interesting insight are made, particularly with the miRNAs that are regulated and may be involved in controlling grape bud dormancy.
Overall, I have a general positive impression of the manuscript and believe it is well written and presents the results well. I do though, have some minor concerns that the authors should consider prior to publication.
- The results presented include a total of 1,336 differentially expression genes, of which more than 65% are modestly differentially regulated (less than 3-fold, 1.58 log2 fold). It would strengthen the manuscript if a subset of differentially expressed genes (i.e. 10) were independently validated using qRT-PCR or northern blot hybridization. This could show that the differential expression is reproducible – at least for a representative subset of induced and repressed genes. Alternatively, the transcriptome dataset could be compared to prior published expression data that is available (Mathiason et al Funct. Int. Genomice 9:81-96, 2009; Fennel et al. Front. Plant Sci. 6:834, 2015; Khalil-Ur-Rehman et al Front. Plant Sci. 8:1340, 2017). This comparison analysis would also potentially support that the observed differential expression in this dataset is reliable/reproducible. Fortunately, this type of analysis appears to have been done for a subset of the differentially expressed miRNAs (Table S5).
- Figure 2 presents a heat map displaying the hormone signaling related gene expression where each biological rep is in a separate column. Although the two sample types (LD buds vs SD buds) cluster together and are clearly separated from each other, I found the variation between the sample-type replicates surprisingly high and somewhat disconcerting. For example, many genes (individual rows in the figure) are not reproducibly repressed in the LD samples where the gene is repressed in one replicate but induced or not differentially expressed in one of the other 2 replicates. Thus, the expression for the LD samples appears highly variable for this select set of hormone related genes. The SD samples shown in the figure are more consistent with one another, but there are also examples when that is not the case there too. This attribute of Figure 2 made me wonder about the variation between the biological reps as a whole (within the entire set of 1336 genes). Unfortunately, Supplementary Table 1 presents only the average FPKM for the SD and LD samples (not the observed results for each biological replicate). That Table should be updated to include the measured FPKM values for each of the reps so the reproducibility between the replicates can be directly evaluated by the readers.
A few other minor comments:
I did not appear to have access to supplementary Figure S1 and S2, so they were not reviewed (although I did have access to Table S1 -S9).
Figure 6 was interesting, but as it was displayed in the PDF it was too small to read and interpret the results. I would recommend presenting this network in a much larger format (possibly as a supplementary figure, to allow the reader to be able to fully interpret it).
There are grammar, wording issues or redundancy in the text that needs review and revision by the authors (see lines 83-84, line 88, lines 258-261 (Fig 6 legend) and line 289)
Author Response
Reviewer #2
Thank you for taking the time to provide comments on the manuscript. Please see point-by-point responses below.
- The results presented include a total of 1,336 differentially expression genes, of which more than 65% are modestly differentially regulated (less than 3-fold, 1.58 log2 fold). It would strengthen the manuscript if a subset of differentially expressed genes (i.e. 10) were independently validated using qRT-PCR or northern blot hybridization. This could show that the differential expression is reproducible – at least for a representative subset of induced and repressed genes. Alternatively, the transcriptome dataset could be compared to prior published expression data that is available (Mathiason et al Funct. Int. Genomice 9:81-96, 2009; Fennel et al. Front. Plant Sci. 6:834, 2015; Khalil-Ur-Rehman et al Front. Plant Sci. 8:1340, 2017). This comparison analysis would also potentially support that the observed differential expression in this dataset is reliable/reproducible. Fortunately, this type of analysis appears to have been done for a subset of the differentially expressed miRNAs (Table S5).
We have compared the RNAseq data to an earlier microarray study with similar treatments (Fennell et al. 2015). There was a strong correlation between the DEGs of the two studies (R2 of 0.8). There were not comparable treatments in the Mathiason et al. or Kahlil et al manuscripts. A new figure (Figure S2) has been added to the results for this comparison.
- Figure 2 presents a heat map displaying the hormone signaling related gene expression where each biological rep is in a separate column. Although the two sample types (LD buds vs SD buds) cluster together and are clearly separated from each other, I found the variation between the sample-type replicates surprisingly high and somewhat disconcerting. For example, many genes (individual rows in the figure) are not reproducibly repressed in the LD samples where the gene is repressed in one replicate but induced or not differentially expressed in one of the other 2 replicates. Thus, the expression for the LD samples appears highly variable for this select set of hormone related genes. The SD samples shown in the figure are more consistent with one another, but there are also examples when that is not the case there too. This attribute of Figure 2 made me wonder about the variation between the biological reps as a whole (within the entire set of 1336 genes). Unfortunately, Supplementary Table 1 presents only the average FPKM for the SD and LD samples (not the observed results for each biological replicate). That Table should be updated to include the measured FPKM values for each of the reps so the reproducibility between the replicates can be directly evaluated by the readers.
We agree that there is variation between replications and have added a statement in the results. Note that these are true biological replicates and 10 node positions were collected from multiple vines for each biological replicate. If it were possible to use a single node position for a replicate, it might be possible to reduce the biological variation. The FPKM of all replicates have been added to Table S1 and as indicated in data accessibility the RNAseq data has been uploaded to NCBI. The treatments provide distinct differences (Figure S1). We have compared the RNAseq data with earlier microarray data (studies were conducted in different years) and there is a strong correlation providing additional confidence in the results. We have removed Figure 2 as the statistically significant positive enrichment of the hormone signaling genes are found in the Gene Set Enrichment Analysis (GSEA, Table S2) and the figures 1a and 1b summarize the transciptomic profiles. It is important to note we report systematic changes in gene expression across pathways and are note relying on specific striking expression of genes. Note microarrays are subject to probe bias as there are a limited number of hybridization probes and it is possible that a given probe may not be representative of the whole transcript. In addition, not all transcripts are represented on the microarray. Thus, a perfect match between microarray and RNASeq are not expected; however, tracking the trends of both will be better than doing qPCR on a few genes. It is of note that there is very good correlation (R2 of 0.8) between the microarray data from previous years and the RNAseq data in this study providing strong confidence in the differences between the treatments.